# Adverse Events in 1406 Patients Receiving 13,780 Cycles of Azacitidine within the Austrian Registry of Hypomethylating Agents—A Prospective Cohort Study of the AGMT Study-Group

**DOI:** 10.3390/cancers14102459

**Published:** 2022-05-17

**Authors:** Michael Leisch, Michael Pfeilstöcker, Reinhard Stauder, Sonja Heibl, Heinz Sill, Michael Girschikofsky, Margarete Stampfl-Mattersberger, Christoph Tinchon, Bernd Hartmann, Andreas Petzer, Martin Schreder, David Kiesl, Sonia Vallet, Alexander Egle, Thomas Melchardt, Gudrun Piringer, Armin Zebisch, Sigrid Machherndl-Spandl, Dominik Wolf, Felix Keil, Manuel Drost, Richard Greil, Lisa Pleyer

**Affiliations:** 13rd Medical Department with Hematology, Medical Oncology, Rheumatology and Infectiology, Paracelsus Medical University, 5020 Salzburg, Austria; m.leisch@salk.at (M.L.); a.egle@salk.at (A.E.); t.melchardt@salk.at (T.M.); r.greil@salk.at (R.G.); 2Salzburg Cancer Research Institute (SCRI) Center for Clinical Cancer and Immunology Trials (CCCIT), Cancer Cluster Salzburg (CCS), 5020 Salzburg, Austria; 3Austrian Group of Medical Tumor Therapy (AGMT) Study Group, 1140 Vienna, Austria; michael.pfeilstoecker@oegkk.at (M.P.); reinhard.stauder@i-med.ac.at (R.S.); sonja.heibl@klinikum-wegr.at (S.H.); heinz.sill@medunigraz.at (H.S.); michael.girschikofsky@ordensklinikum.at (M.G.); margarete.stampfl-mattersberger@gesundheitsverbund.at (M.S.-M.); christoph.tinchon@kages.at (C.T.); bernd.hartmann@lkhf.at (B.H.); andreas.petzer@ordensklinikum.at (A.P.); martin.schreder@wienkav.at (M.S.); david.kiesl@kepleruniklinikum.at (D.K.); sonia.vallet@krems.lknoe.at (S.V.); gudrun.piringer@klinikum-wegr.at (G.P.); armin.zebisch@medunigraz.at (A.Z.); sigrid.machherndl-spandl@ordensklinikum.at (S.M.-S.); dominik.wolf@i-med.ac.at (D.W.); felix.keil@oegk.at (F.K.); 43rd Medical Department for Haematology and Oncology, Hanusch Hospital, 1140 Vienna, Austria; 5Department of Internal Medicine V, Innsbruck Medical University, 6020 Innsbruck, Austria; 64th Medical Department of Internal Medicine, Hematology, Internistic Oncology and Palliative Medicine, Klinikum Wels-Grieskirchen GmbH, 4600 Wels, Austria; 7Division of Hematology, Medical University of Graz, 8036 Graz, Austria; 81st Medical Department, Hematology with Stem Cell Transplantation, Hemostaseology and Medical Oncology, Ordensklinikum Linz GmbH Elisabethinen, 4020 Linz, Austria; 9Department of Internal Medicine 2, Klinik Donaustadt, 1220 Vienna, Austria; 10Department for Hemato-Oncology, LKH Hochsteiermark, 8700 Leoben, Austria; 11Department of Internal Medicine, Landeskrankenhaus Feldkirch, 6800 Feldkirch, Austria; 12Medical Oncology and Hematology, Internal Medicine I, Ordensklinikum Linz GmbH Barmherzige Schwestern, 4020 Linz, Austria; 131st Department of Internal Medicine, Center for Oncology and Hematology, Klinik Ottakring, 1160 Vienna, Austria; 14Department of Hematology and Medical Oncology, Kepleruniversitätsklinikum, 4020 Linz, Austria; 15Department of Internal Medicine 2, University Hospital Krems, Karl Landsteiner Private University of Health Sciences, 3500 Krems, Austria; 16Otto Loewi Research Center for Vascular Biology, Immunology and Inflammation, Division of Pharmacology, Medical University of Graz, 8036 Graz, Austria; 17Assign Data Management and Biostatistics GmbH, 6020 Innsbruck, Austria; manuel.drost@assigndmb.com

**Keywords:** azacitidine, treatment, acute myeloid leukemia, myelodysplastic syndromes, chronic myelomonocytic leukemia, adverse events, toxicity, real-world evidence, prospective cohort study

## Abstract

**Simple Summary:**

Azacitidine is thus far the only drug shown to prolong overall survival and is, therefore, the recommended (backbone) treatment in patients diagnosed with myelodysplastic syndromes, chronic myelomonocytic leukemia and acute myeloid leukemia who are not eligible for intensive chemotherapy. Detailed reports on adverse events are often lacking. We performed a thorough analysis of the adverse events that occur during treatment with azacitidine in the largest cohort of patients treated with this drug published so far. We also compared the frequency of adverse events documented in our cohort to published data from randomized clinical trials with an azacitidine monotherapy arm. Adverse event documentation in the Austrian Registry was high. Hematologic adverse events occurred at a similar rate compared to published trials, whereas gastrointestinal toxicities were significantly less commonly reported. Our data complement results from clinical trials with real-world evidence and form a reference for future combination strategies with azacitidine.

**Abstract:**

**Background:** Azacitidine is the treatment backbone for patients with acute myeloid leukemia, myelodysplastic syndromes and chronic myelomonocytic leukemia who are considered unfit for intensive chemotherapy. Detailed reports on adverse events in a real-world setting are lacking. **Aims:** To analyze the frequency of adverse events in the Austrian Registry of Hypomethylating agents. To compare real-world data with that of published randomized clinical trials. **Results:** A total of 1406 patients uniformly treated with a total of 13,780 cycles of azacitidine were analyzed. Hematologic adverse events were the most common adverse events (grade 3–4 anemia 43.4%, grade 3–4 thrombopenia 36.8%, grade 3–4 neutropenia 36.1%). Grade 3–4 anemia was significantly more common in the Registry compared to published trials. Febrile neutropenia occurred in 33.4% of patients and was also more common in the Registry than in published reports. Other commonly reported adverse events included fatigue (33.4%), pain (29.2%), pyrexia (23.5%), and injection site reactions (23.2%). Treatment termination due to an adverse event was rare (5.1%). **Conclusion:** The safety profile of azacitidine in clinical trials is reproducible in a real-world setting. With the use of prophylactic and concomitant medications, adverse events can be mitigated and azacitidine can be safely administered to almost all patients with few treatment discontinuations.

## 1. Introduction

In the last two decades, the hypomethylating agents azacitidine and decitabine have been the mainstay of treatment for myelodysplastic syndromes, acute myeloid leukemia and chronic myelomonocytic leukemia in patients who are not fit for intensive chemotherapy and/or allogeneic bone marrow transplantation. Myelodysplastic syndromes, acute myeloid leukemia and chronic myelomonocytic leukemia comprise a spectrum of myeloid malignancies, characterized by hematopoietic insufficiency and expansion of malignant bone marrow blasts [1,2]. While they are classified as separate disease entities, they share many clinical features (i.e., dysplasia, cytopenia, transfusion dependence and infections as the most common cause of death amongst others). Approximately one third of patients with myelodysplastic syndromes and chronic myelomonocytic leukemia transform to acute myeloid leukemia, and as such may represent a disease continuum with differing prognoses along the trajectory. In addition, these diseases are often treated similarly.

Based on superior efficacy and significantly prolonged overall survival compared to conventional care regimens, azacitidine is approved for the treatment of patients who are ineligible for allogeneic bone marrow transplantation and are diagnosed with (i) acute myeloid leukemia with >30% marrow blasts, (ii) higher-risk myelodysplastic syndromes according to the International Prognostic Scoring System (IPSS; comprises intermediate-2 and high-risk groups), or (iii) myelodysplastic chronic myelomonocytic leukemia with a white blood cell count of <13.0 G/L and with ≥10% bone marrow blasts by the European medicine agency (EMA) [3,4]. Azacitidine is approved for the treatment of all patients with myelodysplastic syndromes and chronic myelomonocytic leukemia, but only for patients with low blast count acute myeloid leukemia (bone marrow blasts 20–30%) by the FDA [5]. The efficacy of azactidine has also been shown in a real-world setting [6,7,8,9,10,11,12,13,14,15,16,17,18,19].

In patients with chronic myelomonocytic leukemia, approval was based on 6–14 patients included in myelodysplastic syndromes trials [3,20]. Decitabine has recently been compared to hydroxyurea in a randomized phase 3 trial in patients with myeloproliferative chronic myelomonocytic leukemia. In this study, decitabine did not result in an overall survival benefit compared to hydroxyurea, but more patients in the decitabine group were able to proceed to allogeneic transplantation [21]. Our group has recently published the largest retrospective analysis, including 949 patients with chronic myelomonocytic leukemia, comparing hypomethylating agents (azacitidine in 84% of the patients) to other treatment regimens (intensive chemotherapy, hydroxyurea or allogeneic transplantation). We show that hypomethylating agents are associated with superior outcomes compared to other regimens in patients with higher risk chronic myelomonocytic leukemia, underlining its efficacy in this disease [22].

Despite the approval of several new drugs (i.e., gilterinib, ivosidenib, enasidenib, glasdegib, CPX-351, oral azacitidine, luspatercept amongst others) in recent years [23], azacitidine has been and continues to be the backbone for combination strategies with new substances in numerous clinical trials. Most notably, azacitidine is now approved in combination with venetoclax for patients with acute myeloid leukemia who are considered unfit for intensive chemotherapy, due to comorbities based on the OS benefit over azacitidine monotherapy in the VIALE-A trial [24].

Treatment with azacitidine is generally well tolerated and most adverse events are hematologic in nature [3,4,24,25,26]. Nevertheless, there are side effects after treatment with azacitidine and patient numbers in the pivotal phase 3 trials were relatively small, ranging from 140 to 236 patients [3,4,24,25,26]. In this paper, we report a detailed analysis of the Austrian Registry of Hypomethylating agents, focusing on adverse events during treatment with azacitidine in a period covering more than a decade in the largest real-world cohort of patients with acute myeloid leukemia, myelodysplastic syndromes and chronic myelomonocytic leukemia to date. In addition, the frequency of adverse events occurring in the Austrian Registry of Hypomethylating Agents was compared with those occurring in the azacitidine treatment arms of randomized clinical trials.

## 2. Methods

The Austrian Registry of Hypomethylating Agents of the Austrian Group of Medical tumor Therapy (AGMT) (NCT01595295; registered May 2012) is a multicenter database that includes patients with acute myeloid leukemia, myelodysplastic syndromes and chronic myelomonocytic leukemia, who were treated with hypomethylating agents during the course of their disease. Before 2014, only patients treated with azacitidine were included and the Registry was termed the Austrian Azacitidine Registry before 2014. This Registry adheres to published quality guidelines of the U.S. Department of Health and Human Services Agency for Healthcare Research and Quality. All the patients alive at the time of inclusion in the Registry had to sign an informed consent.

Between February 2009 and April 2021, patients from 14 specialized centers for hematology and medical oncology in Austria were included. The data cleaning date was 1 April 2021. The sole inclusion criteria were the diagnosis of acute myeloid leukemia, myelodysplastic syndromes or chronic myelomonocytic leukemia according to WHO criteria and treatment with at least one dose of azacitidine. No formal exclusion criteria existed, as the aim was to include all patients treated with azacitidine, irrespective of age, comorbidities, and/or number of previous lines of treatment. Informed consent to allow the collection of personal data was obtained for all the retrospectively documented patients who were alive, as well as for all the prospectively included patients.

Registry design, data collection and monitoring, as well as assessment of efficacy, safety and endpoints within the Austrian Registry were performed as previously described [7].

Participation in this Registry did not exempt the participating center from their legal reporting obligations. The individual participating centers were instructed to report adverse reactions to the concerned competent authorities, following regulations in the current or future version of Austrian legislation. The final study reported will be uploaded to the Austrian regulatory authorities by the AGMT Study Group (www.basg.gv.at).

Treatment emergent hematologic adverse events were calculated based on differential blood count values and transfusion (in)dependence entered into the electronic case report form at day 1 of every azacitidine treatment cycle, according to the Common Terminology Criteria for Adverse Events version 5.0 (CTCAE v5.0) [27], which are depicted in the Appendix A. Additionally, the values for creatinine, aspartate transaminase (AST or GOT), alanine transaminase (ALT or GPT) and bilirubin were entered at every treatment cycle and were used to calculate the laboratory adverse events according to CTCAE v5.0 (Appendix A). Non-hematologic adverse events were reported and graded by the treating physician according to CTCAE version 5.0. The total number of adverse events, as well as the number of patients experiencing an adverse event, were reported. If the same adverse event occurred more than once in the same patient, the worst grade was used in this analysis. Overall survival was defined as the time from the first day of treatment to death from any cause. The response to azacitidine was assessed according to the European leukemia net (ELN) criteria for acute myeloid leukemia [28] and according to the International working group (IWG) criteria for myelodysplastic syndromes and chronic myelomonocytic leukemia [29].

### Statistics

Statistical analyses were performed by Assign Data Management and Biostatistics GmbH with SAS^®^ 9.4, and by ML with IBM-SPSS statistics v27. Chi-squared tests were used for categorical variables and Wilcoxon tests for continuous variables. The results were reported as significant when *p* < 0.05.

## 3. Results

### 3.1. Baseline Characteristics of Patients Included in the Austrian Registry

A total of 1519 patients were included in the Austrian Registry of Hypomethylating Agents. For this analysis, 67 patients were excluded as they had received decitabine. The baseline characteristics and adverse events of the decitabine cohort are shown in Appendix A. A further 46 patients were excluded due to insufficient follow up data, leaving 1406 patients for this analysis (Figure 1.. At the time of treatment start with azacytidine, 504 patients had a diagnosis of myelodysplastic syndromes, 133 had chronic myelomonocytic leukemia and 769 had acute myeloid leukemia, respectively. The median age of the total cohort was 73 years (IQR 67.0–78.0) and 549 patients (39%) were female. Most patients had an Eastern Cooperative Oncology Group (ECOG) performance score of 0–1 (*n* = 1073; 76.3%). Further baseline variables of the total cohort and of the patients stratified by diagnosis at azacitidine treatment start are listed in Table 1. Data regarding the mutational landscape of the study cohort were available in 173 (12.3%) patients. The patients had a median of one mutation (IQR 1–2). The most common mutations included mutations in NPM1 (*n* = 66), FLT3 (*n* = 47), NF1 (*n* = 27) and TET2 (*n* = 22), respectively.

A total of 528 (37.6%) out of 1406 patients had cardiac comorbidities, 259 (18.4%) had diabetes and 238 (16.9%) had renal impairment at the time of treatment start with azacytidine, resulting in an HCT-CI score of ≥3 in 470 (33.4%) patients. Other comorbidities are listed in Appendix A.

### 3.2. Treatment Characteristics and Treatment Outcomes

The median follow-up (interquartile range (IQR)) from azacitidine start was 10.7 (4.1–21.2) months. A total of 13.780 cycles were documented, amounting to 1514 treatment years. The median (IQR) treatment duration with azacitidine was 5.1 months (1.9–12.1), corresponding to a median (IQR) of five (2–12) treatment cycles. Azacitidine was used as the 1st line treatment in 838 patients (59.6%), as the 2nd line treatment in 301 patients (21.4%), and as the ≥3rd line treatment in 267 (19.0%) of 1406 patients, respectively.

Out of 1406 treated patients, 639 (45.4%) had an objective response, with 154 (11.0%) patients achieving a complete remission (CR), 54 (3.8%) a complete remission with incomplete marrow recovery (CRi), 84 (6.0%) a morphologic leukemia free state (MLFS), 25 (1.8%) a partial remission (PR) and 322 (22.9%) a hematologic improvement (HI) as the best response, respectively.

The median (IQR) overall survival was 9.7 (3.8–18.8) months. The 1- and 3-year survival rates after azacitidine start were 49.2% and 17.9% for the total cohort, respectively. Further treatment characteristics and outcomes are listed in Appendix A.

### 3.3. Documented Adverse Events in the Austrian Registry

A total of 16023 adverse events were documented in 13780 cycles of azacitidine, with 8341 (52.0%) adverse events being grade 1–2 and 6275 (39.1%) being grade 3–4, respectively. Information about the adverse event grade was missing in 1407 (8.7%) adverse events. Of 1406 total patients, 1083 (77.0%) experienced at least one documented adverse event, with 749 (53.2%) experiencing at least one grade 3–4 adverse event.

The most common lower grade (grade 1–2) adverse events were fatigue (*n* = 433, 30.0%), pain (*n* = 368, 26.2%), injection site reactions (*n* = 316, 22.5%) and pyrexia (*n* = 287; 20.4%) (Table 2). Grade 1–2 pyrexia was reported more often in patients with acute myeloid leukemia (23.7%), as compared to patients with myelodysplastic syndromes (18.3%) or chronic myelomonocytic leukemia (9.8%), respectively (*p* = 0.0003). Gastrointestinal toxicity was usually of grade 1–2 with nausea (*n* = 137; 9.7%), diarrhea (*n* = 129; 9.2%) and constipation (n = 117; 8.3%) being the most commonly reported gastrointestinal adverse events. Other documented adverse events are listed in Table 2.

The most commonly documented higher grade (grade 3–4) adverse event was febrile neutropenia, which was reported in 470 (33.4%) of 1406 patients. Febrile neutropenia occurred significantly more often in patients with myelodysplastic syndromes (38.9%) and acute myeloid leukemia (32.5%) than in patients with chronic myelomonocytic leukemia (18.0%) (*p* ≤ 0.0001). Other grade 3–4 adverse events included pneumonia (*n* = 80; 5.7%), fatigue (*n* = 48; 3.4%) and pyrexia (*n* = 43; 3.1%) (Table 2).

The cumulative effects of 122 adverse events (0.7%) that occurred in 16 (0.09%) patients resulted in ICU admission or were classified as life threatening. The cumulative effect of 256 adverse events (1.6%) led to a fatal outcome in 33 (2.3%) of 1406 patients. The fatal adverse events occurred after a median of 3 (IQR 2–8) azacitidine treatment cycles. Ten out of thirty-three patients (30.3%) with a fatal adverse event experienced the adverse event in cycle 1–2 or >7. At the timepoint of the fatal adverse event, the patients had a median of 6 (IQR 4–10) adverse events. The most common were febrile neutropenia (32 adverse events total), pneumonia (24 adverse events) and sepsis (18 adverse events).

The adverse event duration was <3 days in 2677 (16.7%), 3–6 days in 4770 (29.7%), 1–2 weeks in 3840 (23.9%), 2–3 weeks in 1593 (9.9%), 3–4 weeks in 914 (5.7%), and >4 weeks in 2210 (13.8%) adverse events out of the 16023 documented adverse events, respectively. Grade 3–4 adverse events resolved within <3 days in 718 (11.4%), 3–6 days in 1606 (25.5%), 1–2 weeks in 1540 (24.5%), 2–3 weeks in 757 (12.0%), 3–4 weeks in 441 (7.0%), and >4 weeks in 1203 (19.1%) of the documented grade 3–4 adverse events, respectively. The most common adverse events lasting longer than 4 weeks were thrombopenia (391 events), febrile neutropenia (343 events), anemia (286 events) and fatigue (238 events), respectively.

Most documented adverse events occurred within the first four azacitidine treatment cycles. A total of 779 (55.4%) out of 1406 patients experienced an adverse event of grade 1–4 in cycle one, whereas 588 (48.9%) of 1203, 375 (35.9%) of 1045 and 328 (35.3%) of 929 patients experienced an adverse event of grade 1–4 in cycles two, three and four, respectively. The adverse event frequency remained in the 20–30% range from cycle five onward. The same trend was observed for adverse events of grade 1–2 and grade 3–4, respectively (Figure 2).

Hematologic adverse events were more common in patients who received azacitidine as a 1st line treatment than in patients who received azacitidine in later treatment lines. As such, grade 3–4 anemia was observed in 401 (47.8%) of 838 1st line patients and 209 (36.7%) of 568 ≥ 2nd line treated patients, respectively (*p* = 0.0004). Grade 3–4 neutropenia was observed in 365 (43.5%) 1st line patients, as compared to 143 (25.1%) ≥ 2nd line patients (*p* ≤ 0.00001) and grade 3–4 thrombopenia was observed in 340 (40.5%) 1st line patients, as compared to 177 (31.1%) ≥ 2nd line patients, respectively (*p* = 0.0003). There was no difference in non-hematologic and infectious adverse events between the 1st and later line treated patients (data not shown).

In total, 3509 (21.8%) of 16,023 documented adverse events, occurring in 459 (32.6%) of 1406 patients, were considered to be associated with the use of azacitidine. Adverse events directly attributable to azacitidine most commonly included injection site reactions (n = 214, 15.2%), febrile neutropenia (*n* = 152, 10.8%), thrombopenia (*n* = 128, 9.1%) and fatigue (*n* = 88, 6.3%) (data not shown).

### 3.4. Calculated Treatment-Emergent Adverse Events in the Austrian Registry

In addition to the documented adverse events described above, treatment emergent adverse events (TEAEs) were calculated from data entered into the eCRF at the start of each azacitidine treatment cycle. Differential blood count values and the number of required transfusions were documented at the start of each azacitidine treatment cycle, thus, enabling the calculation of treatment-emergent hematologic adverse events for each cycle. Grade 3–4 anemia, thrombopenia and neutropenia were observed in 610 (43.4%), 517 (36.8%) and 508 (36.1%) of 1406 patients, respectively (Table 2). Grade 3–4 anemia was observed more frequently in patients with myelodysplastic syndromes (*n* = 241, 47.8%) and chronic myelomonocytic leukemia (*n* = 64, 48.1%), as compared to patients with acute myeloid leukemia (*n* = 305, 39.7%) (*p* = 0.009) (Table 2). Similarly, grade 3–4 neutropenia was observed most frequently in patients with myelodysplastic syndromes (*n* = 208, 41.3%), then in patients with chronic myelomonocytic leukemia (*n* = 47, 35.3%) or acute myeloid leukemia (*n* = 253, 32.9%), respectively (*p* = 0.008). Grade 3–4 thrombopenia occurred at a similar rate in the following three subgroups: 38.9%, 33.8% and 35.9% for patients with myelodysplastic syndromes, chronic myelomonocytic leukemia and acute myeloid leukemia, respectively (*p* = 0.1074) (Table 2).

Highest grade (i.e., the highest grade of either anemia, thrombopenia or neutropenia) calculated hematologic adverse event frequency decreased only slightly over time. As such, 1398 of 1406 patients (99.4%) experienced a grade 1–4 hematologic adverse event in cycle 1 compared to 332 of 375 patients (88.5%) in cycle 12, respectively (Figure 3). Grade 3–4 hematologic adverse event frequency decreased over time, with 1133 of 1406 patients (80.6%) experiencing a grade 3–4 hematologic adverse event in cycle one compared to 182 of 375 patients (48.5%) in cycle 12, respectively (Figure 3).

The levels of creatinine, GOT, GPT and bilirubin were assessed at the start of each azacitidine treatment cycle, thus, enabling the calculation of treatment emergent laboratory anomalies for each cycle. Grade 1–2 increases in bilirubin, GOT, GPT and creatinine were observed in 248 (17.6%), 250 (17.7%), 322 (22.9%) and 330 (23.4%) of 1406 patients in the total cohort, respectively (Table 2). Grade 3–4 increases in bilirubin, GOT, GPT and creatinine were reported in 157 (11.1%), 178 (12.6%), 221 (15.7%) and 270 (19.2%) patients in the entire cohort, respectively (Table 2). The increases in GOT and GPT were not significantly different in the myelodysplastic syndrome, chronic myelomonocytic leukemia and acute myeloid leukemia subgroups. The bilirubin increases were significantly more common in patients with MDS (*n* = 205, 40.6%) and CMML (*n* = 46, 34.5%) compared to AML (*n* = 154, 19.9%) patients, respectively (*p* ≤ 0.0001). The increases in creatinine were significantly different in the three subgroups and occurred most commonly in CMML (*n* = 97, 72.4%) followed by MDS (*n* = 257, 50.9%) and AML patients (*n* = 246, 31.9%) (*p* ≤ 0.0001; Table 2).

### 3.5. Documented Infections in the Austrian Registry

A total of 3735 infectious events were documented, 2215 (59.2%) of which were grade 3–4. At least one infectious event occurred in 1241 (88.2%) of 1406 patients, respectively.

A pathogen was identified in 1570 (42.0%) of 3735 infectious events, with 1069 (28.6%) being bacterial, 211 (5.6%) viral, 72 (1.9%) fungal and 218 (5.8%) a combination of more than one pathogen, respectively (data not shown). Pneumonia was the most common reported infection and occurred in 287 (20.4%) of 1406 patients, with 80 patients (5.7%) experiencing grade 3–4 pneumonia. Upper respiratory tract infections occurred in 240 (17.1%) of 1406 patients, with 13 (0.9%) patients experiencing grade 3–4 upper respiratory tract infections. Other infectious adverse events are listed in Appendix A.

### 3.6. Impact of Adverse Events on Azacitidine Treatment

Adverse events resulted in azacitidine treatment modifications in 669 (47.6%) of 1406 patients, with prolongation of cycle duration (i.e., longer than 28 days), treatment interruptions, dose reductions and termination of azacitidine treatment occurring in 294 (20.9%), 209 (14.8%), 125 (8.9%), and 73 (5.1%) of 1406 patients, respectively (data not shown).

### 3.7. Treatment and Outcome of Adverse Events

The treatment of an adverse event was necessary for 11.593 (72.3%) out of 16.023 total documented adverse events in 986 (70.1%) of 1406 patients, respectively. Of the 3735 reported infectious events, 3048 (81.6%) were treated with anti-infectious agents; intravenous antibiotics, oral antibiotics, antiviral agents and antifungals were used in 1652 (44.2%), 1221 (32.7%), 117 (3.1%) and 58 (1.6%) of events, respectively.

Hospitalization was required for 5503 (34.3%) of 16.023 adverse events and in 620 (44.0%) of 1406 patients, respectively.

### 3.8. Comparison of Adverse Event Frequency in Patients with Myelodysplastic Syndromes or Chronic Myelomonocytic Leukemia with Data from Clinical Trials

We aimed to compare the frequency of adverse events in the Austrian Registry of Hypomethylating Agents with published clinical trials. The following search terms were used in the PubMed database to identify relevant clinical trials: “myelodysplastic syndrome”, “MDS”, “chronic myelomonocytic leukemia”, “CMML” in combination with “azacitidine”. In addition, the filter “clinical trial” was used. Fifty-two trials were identified, of which fifty were in the English language. Of these, *n* = 14 included an azacitidine monotherapy arm. After exclusion of clinical trials with less than fifty patients in the azacitidine arm, six trials remained. Of these, only 3 trials including 220, 175 and 177 patients reported detailed adverse event information, thus, qualifying for this study (Figure 1, Table 3) [3,26,31].

For this analysis, the patients with chronic myelomonocytic leukemia and myelodysplastic syndromes were grouped together, since clinical trials performed in patients with myelodysplastic syndromes mostly also included patients with chronic myelomonocytic leukemia. The adverse event frequency of CALGB trials 8291 and 9221 are summarized in detail in the FDA, prescribing information for Vidaza^®^. Thus, the respective data were taken from the prescribing information but are referenced as CALGB trials in the tables and hereafter.

There were relevant differences regarding the baseline and prognostic factors between the Austrian Registry and the clinical trials. Gender distribution was significantly different across all the trials. The patients in the Austrian Registry had a significantly higher ECOG performance score, whereas IPSS prognostic score was higher in the clinical trials (Table 3).

The rate of grade 1–4 neutropenia was significantly more frequent than in the AZA-MDS-001 trial (65.7%) and in the Austrian Registry of Hypomethylating Agents (48.5%), than in the CALGB- (32.3%) and SUPPORT (26.0%) trials (*p*< 0.0001; Table 3), with similar results being observed for grade 3–4 neutropenia, respectively. Grade 1–4 anemia was significantly more frequent in the CALGB trials (69.5%) and the Austrian Registry (62.5%), than in the AZA-MDS-001 (51.4%) and SUPPORT (14.7%) trials, respectively (*p* < 0.0001; Table 3). Grade 3–4 anemia was significantly more common in the Austrian Registry of Hypomethylating Agents (47.9%) than in the AZA-MDS-001 (13.7%) and SUPPORT (11.3%) trials (*p* ≤ 0.0001), respectively (Table 3). The rate of pyrexia was significantly higher in the CALGB trial than in all the other reports (51.8 vs. 19.0–30.3%; *p* < 0.0001) and the rate of febrile neutropenia was highest (34.5%) in the Austrian cohort, compared to 13–21% in the other reports (*p* < 0.0001). Grade 1–2 (but not grade 3–4) pneumonia and grade 1–2 (but not grade 3–4) upper respiratory tract infections were reported significantly more often in the Austrian Registry of Hypomethylating Agents than in the CALGB and AZA-MDS-001 trials, whereas the rate of urinary tract infections grade 1–4 was similar to that observed in the AZA-MDS-001 trial (8.7 vs. 8.9%, *p* = 0.8765; Table 3).

Gastrointestinal toxicity, including grade 1–4 nausea, grade 1–4 diarrhea and grade 1–4 constipation, was documented significantly less frequently in the Austrian Registry of Hypomethylating Agents compared to the other reports (*p* ≤ 0.0001) (Table 3).

Injection site reactions were reported more often in the AZA-MDS-001 trial (29.1%) and the Austrian Registry (25.9%) than in the CALGB trials (13.6%) (*p* = 0.0002). Fatigue grade 1–2 (but not grade 3–4) was more common in the Austrian cohort (34.4%) than in the the other reports (14.1–24.0%) (*p* < 0.0001) (Table 3).

The rate of treatment discontinuation due to an adverse event was highest in the SUPPORT trial (13.5%) and similar in the AZA-MDS-001 trial and the Austrian Registry (5.0 and 5.1%, respectively) (*p* = 0.0001).

The following adverse events could not be compared as they were not reported in the clinical trials: skin/mucosal infection, pain, bilirubin increase, GOT increase, GPT increase and creatinine increase.

### 3.9. Comparison of Adverse Event Frequency in Patients with Acute Myeloid Leukemia with Data from Clinical Trials

Similar to the myelodysplastic syndrome and chronic myelomonocytic leukemia trials, the PubMed database was used to identify relevant clinical trials for acute myeloid leukemia using the search terms “acute myeloid leukemia” or “AML”, in combination with “azacitidine”. In addition, the filter “clinical trial” was used. In total, 60 trials were identified, all of which were in the English language. Of these, *n* = 4 included an azacitidine monotherapy arm. Of these, only 2 trials including 236 and 145 patients reported detailed adverse event information, thus, qualifying for this study (Figure 1, Table 4) [24,25].

There were relevant differences regarding the baseline and prognostic factors between the Austrian Registry and the clinical trials. The patients in the Austrian Registry and the VIALE-A trial had a significantly higher ECOG performance score than the patients in the AZA-AML-001 trial. MRC cytogenetic risk was higher in the clinical trials than in the Austrian Registry (Table 4).

In acute myeloid leukemia patients, the rate of grade 1–4 neutropenia (*p* = 0.0140), grade 1–4 anemia (*p* < 0.0001) and grade 1–4 thrombopenia (*p* < 0.0001) was significantly higher in the Austrian cohort compared to the AZA-AML-001 and VIALE-A trials (Table 4). Febrile neutropenia occurred at a similar rate in the AZA-AML-001 trial (32.2%) and the Austrian Registry of Hypomethylating Agents (32.5%), but was significantly less common in the VIALE A trial (18.8%) (*p* = 0.003).

Pneumonia grade 1–4 occurred at a similar frequency in all the cohorts analyzed (*p* = 0.2800), but grade 3–4 pneumonia was documented less frequently in the Austrian cohort (*p* < 0.0001; Table 4).

Similar to the observations made between the myelodysplastic syndrome cohorts, grade 1–4 gastrointestinal toxicity was significantly less commonly reported (*p* < 0.0001) and fatigue grade 1–4 was reported significantly more often (*p* = 0.0050) in the Austrian Registry, as compared to the AZA-AML-001 and VIALE A trials, respectively (Table 4).

The rate of treatment discontinuation due to an adverse event was highest in the AZA-AML-001 trial (37.0%) and similar in the VIALE-A-trial and the Austrian Registry (3.4 and 5.2%, respectively) (*p* = 0.0001).

The following adverse events could not be compared as they were not reported in the clinical trials: skin/mucosal infection, pain, bilirubin increase, GOT increase, GPT increase, and creatinine increase.

## 4. Discussion

Herein, we report on the frequency and severity of adverse events during treatment with azacitidine in the largest real-world cohort (*n* = 1406) published so far, over an observation period of 14 years and 13,780 applied azacitidine treatment cycles.

The baseline characteristics of the Austrian cohort resemble those of other reports [8,9,10,11,13,14,15,16,17,18,19,32,33] and phase III clinical trials with a median age above 70 years and frequent comorbidities [3,4,26,31,34].

Hematologic adverse events were the most common treatment emergent adverse events, which is in line with the safety profile of azacitidine reported previously [3,4,24,25,26]. The rate of grade 3–4 neutropenia was 36.1% in the Austrian Registry, which lies within the range reported in clinical trials (26.0–61.1%). Similarly, the rate of grade 3–4 thrombopenia (36.8%) in the Austrian Registry was within the range reported in previous reports (23.7–58.3%). However, grade 3–4 anemia was observed significantly more often in the Austrian Registry of hypomethylating agents (43.4%) than in the published trials (11.3–20.1%; *p* < 0.0001). One possible explanation for this observation might be the fact that azacitidine was exclusively used as the first line agent in the clinical trials analyzed, whereas 40.4% of the Austrian cohort received the drug as a ≥2nd line treatment. However, we observed higher rates of hematologic toxicity in the 1st line treated patients compared to the ≥2nd line treated patients; therefore, this cannot be the sole explanation. Other possible factors might be the inclusion of patients with higher ECOG scores in the Austrian Registry, higher patient comorbidity or differences in growth factor usage.

In the Austrian cohort, febrile neutropenia was the most common non-hematologic adverse event, occurring in 33% of all patients. Of note, febrile neutropenia occurred less frequently in chronic myelomonocytic leukemia patients compared to acute myeloid leukemia and myelodysplastic syndromes patients. One possible explanation may be that chronic myelomonocytic leukemia patients might be less immunosuppressed compared to acute myeloid leukemia and myelodysplastic syndromes patients, due to the myeloproliferative phenotype observed in a subset of these patients. The frequency of documented grade 3–4 febrile neutropenia, grade 1–4 upper respiratory tract infections and grade 1–4 fatigue was highest in the Austrian cohort, compared to published trials in both the myelodysplastic syndrome and the acute myeloid leukemia cohorts, underlining a high reporting rate within the Registry.

A possible explanation for the higher rate of febrile neutropenia might be the inclusion of patients with more comorbidities. Detailed information on comorbidities was not available in the clinical trial reports. In addition, we report rates of skin and mucosal infections, pain, bilirubin increase, GOT increase, GPT increase, and creatinine increase. These adverse events were not reported in any of the pivotal trials; therefore, this Registry analysis adds additional information to the safety profile of azacitidine.

Infections occurred in 88% of all the patients in the Austrian cohort, with respiratory infections (pneumonia and upper respiratory tract infections) being the most common with a combined rate of 37.6%. Most infections were bacterial (28.6%) and only a minority (1.9%) were fungal. These data indicate that patients and their household contacts should be vaccinated against respiratory pathogens whenever possible/indicated and standard hygiene concepts should be encouraged. These results also support current practice guidelines that do not suggest the routine use of prophylactic antimicrobial agents in all patients with myelodysplastic syndromes, chronic myelomonocytic leukemia and acute myeloid leukemia treated with azacitidine, but state that antimicrobial prophylaxis should be reserved for severely neutropenic patients or patients with additional risk factors (e.g., corticosteroid treatment) [35].

Lower grade adverse events that were frequently reported included injection site reactions as well as gastrointestinal toxicity, consisting of nausea, diarrhea and constipation. Overall, gastrointestinal toxicities were significantly less commonly reported in the Austrian Registry of Hypomethylating Agents, compared to published trials. Premedication with 5HT3 antagonists is standard practice in Austria, which might partially explain this phenomenon. In addition, patients treated with azacitidine in Austria commonly receive prescriptions for antiemetics, as well as for both loperamide and laxatives, with instructions for the dosages and application schedules in case gastrointestinal toxicity occurs. This might have mitigated or eliminated gastrointestinal symptoms, resulting in a lack of the patient suffering enough to remember to mention the adverse event upon questioning several weeks later, at the start of the next azacitidine treatment cycle. These data underline and enforce the commonly used practice in Austria to prescribe prophylactic and bystander medications.

Adverse events during azacitidine treatment resulted in postponement of the next treatment cycle and dose reductions in about 20% and 10% of the patients, respectively. Treatment discontinuation/termination due to an adverse event occurred in only 5% of the patients in the Austrian Registry of Hypomethylating agents. Since survival after treatment discontinuation with azacitidine is short (median 2.3 months [6]), we suggest a delay of the upcoming cycle or (preferably) maintaining the planned schedule with azacitidine dose reductions whenever possible. With this strategy, most patients will be able to continue azacitidine treatment.

Adverse event frequency decreased over the course of treatment and most adverse events occurred in the first four treatment cycles. This is an important aspect when initiating treatment with azacitidine. Patients should be instructed to report adverse events and physicians should schedule regular visits, in order to diagnose and potentially mitigate adverse events occurring in the first treatment cycles. In particular, the need for blood product supply should be checked regularly in the first cycles, depending on the severity of pre-treatment cytopenias. Patients should be informed that the risk of an adverse event is likely to decrease over time, which might increase treatment adherence after experiencing an adverse event.

Grade 5 (fatal) adverse events were reported in 2.3% of all the patients. In these patients, febrile neutropenia, pneumonia and sepsis were most commonly associated with a fatal outcome. We found a median of six adverse events at the time of death, indicating a multifactorial cause of death. The majority of fatal adverse events occurred either early (in the first two cycles) or later (the seventh cycle and beyond). This is in line with clinical experience that patients are at the highest risk of a fatal outcome when the underlying disease is either not yet or no longer controlled. Therefore, it is often hard to discriminate between drug side effects and the underlying disease as the cause of death.

A potential limitation of this study might be that adverse event reporting was based on the review of patient charts; therefore, the underreporting of adverse events not deemed clinically relevant by the treating physician cannot be excluded. However, the fact that we report higher rates of adverse events that might be perceived as less severe (such as grade 1–2 upper respiratory tract infections, grade 1–2 urinary tract infections, fatigue, pain or injection site reactions) clearly demonstrates a high accuracy of adverse event reporting in this real-world Registry. By calculating treatment emergent hematologic and laboratory anomalies, this potential bias was eliminated with regard to these adverse events.

## 5. Conclusions

Overall, the adverse event reporting in the Austrian Registry of Hypomethylating Agents was high and mostly comparable to that of published randomized clinical trials with some differences in frequency, as outlined above. Furthermore, we have found adverse events not documented in published trials that complement the existing clinical trial data. Azacitidine was well tolerated in our cohort and treatment discontinuation due to adverse events was rarely necessary. Hematologic toxicities were the most common adverse events and occurred primarily in the first four treatment cycles. Therefore, regular visits, blood product and potentially growth factor support, as indicated by published guidelines [36], are important for management, especially in the first treatment cycles [5,37].

Expected adverse events, such as gastrointestinal toxicities, including nausea, diarrhea and/or constipation, could be mitigated and significantly reduced by the commonly used practice in Austria for premedication with 5HT3-antagonists and to prescribe prophylactic and bystander medications for anticipated gastrointestinal toxicity.

Since azacitidine is expected to be a frequent combination partner for new emerging substances, it is important to know its safety profile to differentiate the substance specific side effects. This real-world analysis further feeds the requirement of regulatory authorities [38] and leading experts [39,40,41,42] for assessing the generalizability of clinical trial data in daily clinical practice. The current report is in line with previous work from our group [34], where we observed the reproducible efficacy of azacitidine as the first line treatment in AML, by comparing data from the Austrian Registry of Hypomethylating Agents with the phase-III AZA-AML-001 trial [4], demonstrating the high quality and utility of our database. As such, we believe that this report complements results from clinical trials with real-world evidence and can form a reference for future combination strategies with azacitidine.

## Figures and Tables

**Figure 1 cancers-14-02459-f001:**
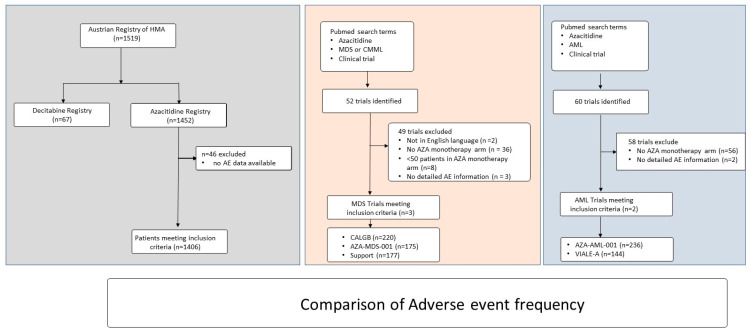
Consort diagram.

**Figure 2 cancers-14-02459-f002:**
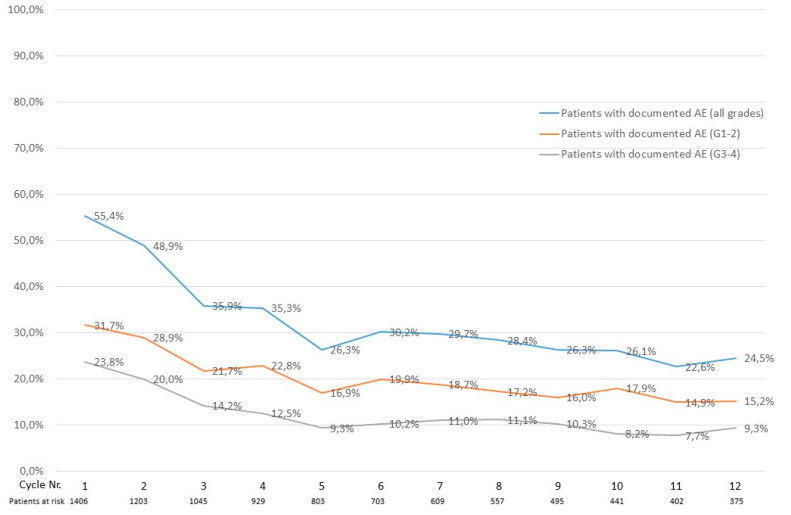
Percentage of patients who experienced a documented AE per azacitidine treatment cycle.

**Figure 3 cancers-14-02459-f003:**
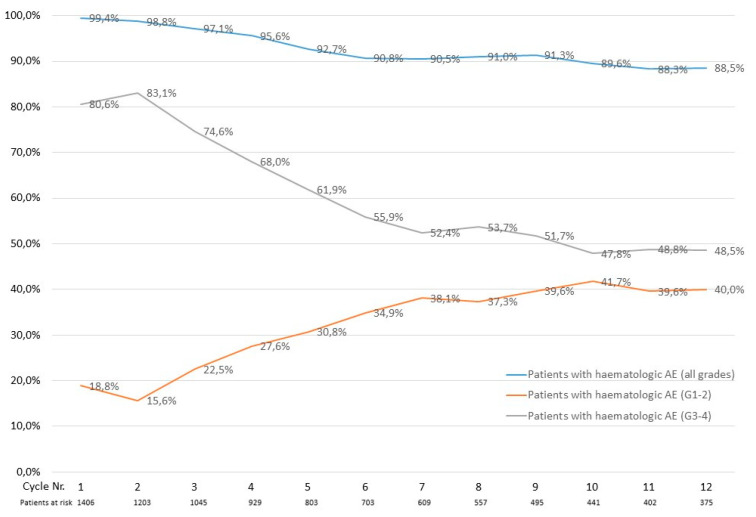
Percentage of patients who experienced a calculated hematologic AE per azacitidine treatment cycle.

**Table 1 cancers-14-02459-t001:** Characteristics of patients included in the Austrian Registry of Hypomethylating Agents at azacitidine treatment start.

	Total Cohort (*n* = 1406)	MDS (*n* = 504)	CMML (*n* = 133)	AML (*n* = 769)	*p* Value
Initial diagnosis: MDS, n (%)	622 (44.2)	470 (93.3)	15 (11.3)	137 (17.8)	NA
CMML	133 (9.5)	4 (0.8)	106 (79.7)	23 (3.0)	
AML ^1^	583 (41.5)	7 (1.4)	1 (0.8)	575 (74.8)	
CMPD	41 (2.9)	6 (1.2)	4 (3.0)	31 (4.0)	
Unknown	27 (1.9)	17 (3.4)	7 (5.3)	3 (0.4)	
Diagnosis at azacitidine start: MDS, n (%)	504 (35.8)	504 (100)	0 (0.0)	0 (0.0)	NA
CMML	133 (9.5)	0 (0.0)	133 (100)	0 (0.0)	
AML^1^	769 (54.7)	0 (0.0)	0 (0.0)	796 (100)	
Unknown	0 (0.0)	0 (0.0)	0 (0.0)	0 (0.0)	
Mean age (SD), years	71.9 (9·88)	71.8 (9.48)	73.0 (7.96)	71.8 (10.42)	0.5375 0.8778
Median (IQR)	73.0 (67.0–78.0)	72.0 (66.0–78.0)	74.0 (69.0–79.0)	73.0 (67.0–79.0)	
Min-max	23.0–99.0	36.0–99.0	38.0–87.0	23.0–93.0	
≥75 years, n (%)	605 (43.0)	216 (42.9)	60 (45.1)	329 (42.8)	
Unknown	0 (0.0)	0 (0.0)	0 (0.0)	0 (0.0)	
Sex: Female, n (%)	549 (39.0)	175 (34.7)	52 (39.1)	322 (41.9)	0.0380
Male	857 (61.0)	329 (65.3)	81 (60.9)	447 (58.1)	
Unknown	0 (0.0)	0 (0.0)	0 (0.0)	0 (0.0)	
ECOG-PS: 0–1, n (%)	1073 (76.3)	404 (80.1)	108 (81.2)	561 (72.9)	0.0121
2–4	333 (23.7)	100 (19.9)	25 (18.8)	208 (27.1)	
Unknown	0 (0.0)	0 (0.0)	0 (0.0)	0 (0.0)	
HCT-CI risk group: Low risk, n (%)	434 (30.9)	168 (33.3)	42 (31.6)	224 (29.1)	0.3392
Intermediate risk	500 (35.6)	180 (35.7)	42 (31.6)	278 (36.2)	
High risk	470 (33.4)	154 (30.6)	49 (36.8)	267 (34.7)	
Unknown	2 (0.1)	2 (0.4)	0 (0.0)	0 (0.0)	
Treatment-related disease: No, n (%)	1212 (86.2)	423 (83.9)	118 (88.7)	671 (87.3)	0.0406
Yes	187 (13.3)	81 (16.1)	11 (8.3)	95 (12.4)	
Unknown	7 (0.5)	0 (0.0)	4 (3.0)	3 (0.4)	
IPSS cytogenetic risk: Good, n (%)	878 (62.4)	323 (64.1)	93 (69.9)	462 (60.1)	0.1855
Intermediate	203 (14.4)	63 (12.5)	21 (15.8)	119 (15.5)	
Poor	151 (10.7)	62 (12.3)	7 (5.3)	82 (10.7)	
Not evaluable	90 (6.4)	27 (5.4)	8 (6.0)	55 (7.2)	
Unknown	84 (6.0)	29 (5.8)	11 (8.3)	51 (6.6)	
R-IPSS cytogenetic risk: Very good, n (%)	33 (2.3)	11 (2.2)	0 (0.0)	22 (2.9)	0.2126
Good	856 (61.0)	318 (63.1)	93 (69.9)	447 (58.1)	
Intermediate	223 (15.9)	70 (13.9)	22 (16.5)	131 (17.0)	
Poor	99 (7.0)	41 (8.1)	6 (4.5)	52 (6.8)	
Very poor	19 (1.4)	8 (1.6)	0 (0.0)	11 (1.4)	
Not evaluable	90 (6.4)	27 (5.4)	8 (6.0)	55 (7.2)	
Unknown	84 (6.0)	29 (5.8)	4 (3.1)	51 (6.6)	
IPSS risk group: Lower-risk ^2^, n (%)	351 (24.9)	208 (41.2)	73 (54.8)	70 (9.1)	<0.0001
Higher-risk ^3^	889 (63.2)	229 (45.4)	47 (35.3)	613 (79.7)	
Unknown	166 (11.8)	67 (13.3)	13 (9.7)	86 (11.2)	
Red blood cell transfusion dependence: Yes, n (%)	821 (58.4)	266 (52.8)	85 (63.9)	470 (61.1)	0.0051
No	585 (41.6)	238 (47.2)	48 (36.1)	299 (38.9)	
Unknown	0 (0.0)	0 (0.0)	0 (0.0)	0 (0.0)	
Platelet transfusion dependence: Yes, n (%)	1115 (79.3)	414 (82.1)	115 (86.5)	586 (76.2)	0.0038
No	291 (20.7)	90 (17.9)	18 (13.5)	183 (23.8)	
Unknown	0 (0.0)	0 (0.0)	0 (0.0)	0 (0.0)	

CMPD indicates chronic myeloproliferative diseases; ECOG-PS, Eastern Cooperative Oncology Group Performance Status; HCT-CI, Hematopoietic Cell Transplantation-specific Comorbidity Index; IPSS, International Prognostic Scoring System; R-IPSS, revised IPSS. ^1^ According to the WHO 2016 classification [1]. ^2^ IPSS lower risk comprises IPSS low and intermediate-1 risk categories [30]. ^3^ IPSS higher risk comprises IPSS intermediate-2 and high-risk categories.

**Table 2 cancers-14-02459-t002:** Adverse events of patients included in the Austrian Registry of Hypomethylating Agents.

	Total Cohort (*n* = 1406)	MDS (*n* = 504)	CMML (*n* = 133)	AML (*n* = 769)	*p* Value
Calculated adverse events ^1,2^					
Neutropenia:1 Grade1–2, n (%)	97 (6.9)	39 (7.7)	15 (11.3)	43 (5.6)253 (32.9)	0.0090
Grade 3–4	508 (36.1)	208 (41.3)	47 (35.3)		
Lymphopenia:1 Grade 1–2, n (%)	719 (51.1)	217 (43.0)	38 (28.6)	464 (60.3)	<0.0001
Grade 3–4	263 (18.7)	163 (32.3)	25 (18.9)	75 (9.8)	
Anemia:1 Grade 1–2, n (%)	247 (17.6)	70 (13.9)	23 (17.3)	154 (20.0)	0.0080
Grade 3–4	610 (43.4)	241 (47.8)	64 (48.1)	305 (39.7)	
Thrombopenia:1 Grade 1–2, n (%)	168 (11.9)	60 (11.9)	24 (18.0)	84 (10.9)	0.1074
Grade 3–4	517 (36.8)	196 (38.9)	45 (33.8)	276 (35.9)	
Bilirubin increase:2 Grade1–2, n (%)	248 (17.6)	124 (24.6)	33 (24.8)	91 (11.8)	<0.0001
Grade 3–4	157 (11.1)	81 (16.0)	13 (9.7)	63 (8.1)	
GOT increase:2 Grade 1–2, n (%)	250 (17.7)	112 (22.2)	24 (18.0)	114 (14.8)	0.1380
Grade 3–4	178 (12.6)	55 (10.9)	20 (15.0)	103 (13.3)	
GPT increase:2 Grade 1–2, n (%)	322 (22.9)	141 (27.9)	21 (15.7)	160 (20.8)	0.0122
Grade 3–4	221 (15.7)	70 (13.8)	16 (12.0)	135 (17.5)	
Creatinine increase:2 Grade 1–2, n (%)	330 (23.4)	148 (29.3)	54 (40.6)	128 (16.6)	<0.0001
Grade 3–4	270 (19.2)	109 (21.6)	43 (32.3)	118 (15.3)	
Documented adverse events					
Pyrexia: Grade 1–2, n (%)	287 (20.4)	92 (18.3)	13 (9.8)	182 (23.7)	0.0003
Grade 3–4	43 (3.1)	15 (3.0)	1 (0.8)	27 (3.5)	
Febrile neutropenia: Grade 1–2, n (%)	0 (0.0)	0 (0.0)	0 (0.0)	0 (0.0)	<0.0001
Grade 3–4	470 (33.4)	196 (38.9)	24 (18.0)	250 (32.5)	
Pneumonia Grade 1–2, n (%)	207 (14.7)	79 (15.7)	15 (11.3)	113 (14.7)	0.4446
Grade 3–4	80 (5.7)	24 (4.8)	4 (3.0)	52 (6.8)	
Upper resp. infection: Grade 1–2, n (%)	227 (16.1)	93 (18.5)	22 (16.5)	112 (14.6)	0.1811
Grade 3–4	13 (0.9)	3 (0.6)	1 (0.8)	9 (1.2)	
Nausea: Grade 1–2, n (%)	137 (9.7)	56 (11.1)	13 (9.8)	68 (8.8)	0.4103
Grade 3–4	3 (0.2)	1 (0.2)	0 (0.0)	2 (0.3)	
Diarrhea: Grade 1–2, n (%)	129 (9.2)	44 (8.7)	13 (9.8)	72 (9.4)	0.9005
Grade 3–4	10 (0.7)	4 (0.8)	2 (1.5)	4 (0.5)	
Constipation: Grade 1–2, n (%)	117 (8.3)	49 (9.7)	13 (9.8)	55 (7.2)	0.2184
Grade 3–4	2 (0.1)	1 (0.2)	0 (0.0)	1 (0.1)	
Urinary tract infection: Grade 1–2, n (%)	106 (7.5)	40 (7.9)	11 (8.3)	55 (7.2)	0.8263
Grade 3–4	12 (0.9)	6 (1.2)	0 (0.0)	6 (0.8)	
Skin/mucosal infection: Grade 1–2, n (%)	102 (7.3)	40 (7.9)	7 (5.3)	55 (7.2)	0.5643
Grade 3–4	14 (1.0)	9 (1.8)	0 (0.0)	5 (0.7)	
Bacterial infection other: Grade 1–2, n (%)	81 (5.8)	22 (4.4)	6 (4.5)	53 (6.9)	0.1350
Grade 3–4	24 (1.7)	12 (2.4)	2 (1.5)	10 (1.3)	
Injection site reaction: Grade 1–2, n (%)	316 (22.5)	132 (26.2)	31 (23.3)	153 (19.9)	0.0304
Grade 3–4	10 (0.7)	2 (0.4)	0 (0.0)	8 (1.0)	
Fatigue: Grade 1–2, n (%)	422 (30.0)	159 (31.5)	45 (33.8)	218 (28.3)	0.2859
Grade 3–4	48 (3.4)	11 (2.2)	4 (3.0)	33 (4.3)	
Pain: Grade 1–2, n (%)	368 (26.2)	133 (26.4)	41 (30.8)	194 (25.2)	0.3947
Grade 3–4	42 (3.0)	9 (1.8)	5 (3.8)	28 (3.6)	

^1^ Adverse events and grading were calculated according to CTCAE v5.0 from differential blood count values and transfusion requirements entered into the electronic case report form at the start of every azacitidine treatment cycle. ^2^ Adverse events and grading were calculated according to CTCAE v5.0 from lab values entered into the electronic case report form at the start of every azacitidine treatment cycle.

**Table 3 cancers-14-02459-t003:** Comparison of adverse events that occurred in patients with myelodysplastic syndromes or chronic myelomonocytic leukemia treated with azacitidine monotherapy within phase III clinical trials or the Austrian Registry of Hypomethylating Agents.

	CALGB [26] ^1^	AZA-MDS-001 [3]	Support [31]	Austrian Registry	*p* Value
Phase	III	III	III	Registry	NA
Year published	2006	2009	2018	2022	NA
Included diagnosis	Newly diagnosed MDS and CMML	Newly diagnosed MDS and CMML ^2^	Newly diagnosed MDS ^3^	Newly diagnosed/RR MDS and CMML	NA
Allowed IPSS risk categories	Low-high	Int 2-high	Int 1-high	Low-high	NA
Allowed pretreatments	None	None	None	No restrictions	NA
Study design	AZA vs. BSC	AZA vs. CCR	AZA +/− eltrombopag	AZA	NA
Total patients in azacitidine arm, n	220	175	177	637	NA
Median age, yrs	67	69	70	73	
Sex: Male, n (%)	107 (48.6)	132 (74)	124 (70)	410 (64.4%)	<0.0001
ECOG-PS: 0–1, n (%)	149 (67.7)	164 (93.7)	177 (100.0) ^4^	512 (80.4)	<0.0001
2–4	24 (10.9)	11 (6.3)	0 (0.0)	125 (19.6)	
Treatment related disease: Yes, n (%)	NR	NR	NR	92 (14.4)	NA
IPSS risk group: Lower risk, n (%)	NR	5 (3.0)	61 (34.0)	281 (44.1)	<0.0001
Higher risk	NR	158 (89.0)	116 (66.0)	276 (43.3)	
IPSS cytogenetic risk: Good risk, n (%)	NR	83 (46.0)	81 (46.0)	416 (65.3)	<0.0001
Intermediate risk	NR	37 (21.0)	39 (22.0)	84 (13.1)	
Poor risk	NR	50 (28.0)	57 (32.0)	69 (10.8)	
Azacitidine treatment cycles, median	NR	9	6	7.25	NA
Q1–Q3	NR	4–15	NR	3.5–16.0	
Median overall survival, months	20	24.5	18.7	13.9	NA
Treatment discount. due to AE: Yes, n (%)	NR	8 (5.0)	24 (13.5)	33 (5.1)	0.0001
Neutropenia: Grade1–4, n (%)	71 (32.3)	115 (65.7)	46 (26.0)	309 (48.5)	<0·0001
Grade 3–4	NR	107 (61.1)	46 (26.0)	255 (40.0)	<0.0001
Anemia: Grade 1–4, n (%)	153 (69.5)	90 (51.4)	26 (14.7)	398 (62.5)	<0·0001
Grade 3–4	NR	24 (13.7)	20 (11.3)	305 (47.9)	<0.0001
Thrombopenia: Grade 1–4, n (%)	144 (65.5)	122 (69.7)	NR	325 (51.0)	<0.0001
Grade 3–4	NR	102 (58.3)	NR	241 (37.8)	<0.0001
Pyrexia: Grade 1–4, n (%)	114 (51.8)	53 (30.3)	46 (26.0)	121 (19.0)	<0·0001
Grade 3–4	NR	8 (4.6)	5 (2.8)	16 (2.5)	0.3579
Febrile neutropenia: Grade 1–4, n (%)	36 (16.4)	24 (13.7)	38 (21.5)	220 (34.5)	<0·0001
Grade 3–4	NR	22 (12.6)	32 (18.1)	220 (34.5)	<0.0001
Pneumonia Grade 1–4, n (%)	24 (10.9)	22 (12.6)	25 (14.1)	122 (19.2)	0·0100
Grade 3–4	NR	18 (10.3)	10 (5.6)	28 (4.4)	0.0115
Upper resp. infection: Grade 1–4, n (%)	28 (12.7)	16 (9.1)	NR	119 (18.7)	0.0033
Grade 3–4	NR	3 (1.7)	NR	4 (0.6)	0.1685
Urinary tract infection: Grade 1–4, n (%)	NR	15 (8.6)	NR	57 (8.9)	0.8765
Grade 3–4	NR	3 (1.7)	NR	6 (0.9)	0.3873
Nausea: Grade 1–4, n (%)	155 (70.5)	84 (48.0)	46 (26.0)	70 (11.0)	<0·0001
Grade 3–4	NR	3 (1.7)	1 (0.6)	1 (0.2)	0.0362
Diarrhea: Grade 1–4, n (%)	80 (36.4)	38 (21.7)	25 (14.1)	63 (9.9)	<0·0001
Grade 3–4	NR	1 (0.6)	1 (0.6)	6 (0.9)	0.8208
Constipation: Grade 1–4, n (%)	74 (33.6)	88 (50.3)	57 (32.2)	63 (9.9)	<0·0001
Grade 3–4	NR	2 (1.1)	2 (1.1)	1 (0.2)	0.1151
Injection site reaction: Grade 1–4, n (%)	30 (13.6)	51 (29.1)	NR	165 (25.9)	0.0002
Grade 3–4	NR	1 (0.6)	NR	2 (0.3)	0.6190
Fatigue: Grade 1–4, n (%)	NR	43 (24.0)	25 (14.1)	219 (34.4)	<0.0001
Grade 3–4	NR	6 (3.4)	1 (0.6)	15 (2.4)	0.1775

NA indicates not applicable; RR, relapsed or refractory; Int, intermediate; AZA, azacitidine; BSC, best supportive care; CCR, conventional care regimen; NR, not reported; ECOG-PS, Eastern Cooperative Oncology Group Performance Score; IPSS, International Prognostic Scoring System. ^1^ As cited in the FDA prescribing information. Includes only patients of CALGB studies (includes cross over patients). ^2^ Included 27 (15.4%) of 175 patients with AML according to WHO criteria. ^3^ Platelet count of <75 G/L. ^4^ Formally not reported inclusion criteria were ECOG 0–2.

**Table 4 cancers-14-02459-t004:** Comparison of adverse events for patients with acute myeloid leukemia treated with azacitidine monotherapy within phase III clinical trials or the Austrian Registry.

	AZA-AML 001 [4]	VIALE A [24]	Austrian Registry	*p* Value
Phase	III	III	Registry	NA
Year published	2015	2020	2022	NA
Included diagnosis	Newly diagnosed AML	Newly diagnosed AML	Newly diagnosed/RR AML	NA
Allowed pretreatments	None	None	No restrictions	NA
Study design	AZA vs.·CCR	AZA +/− venetoclax	AZA	NA
Total patients in azacitidine arm, n	236	145	769	NA
Median age, yrs	75	76	73	NA
Sex: Male, n (%)	139 (57.7)	87 (60)	447 (58.1)	0.9077
ECOG-PS: 0–1, n (%)	236 (100.0)	81 (56.0)	561 (72.9)	<0.0001
2–4	0 (0.0)	64 (44.0)	208 (27.1)	
Treatment related disease: Yes, n (%)	8 (3.3)	9 (6.2)	95 (12.4)	0.0008
MRC cytogenetic risk: Low risk, n (%)	0 (0.0)	0 (0.0)	23 (3.0)	0.0003
Intermediate risk	155 (64.3)	89 (61.0)	481 (62.5)	
Poor risk	85 (35.3)	56 (39.0)	159 (20.7)	
Azacitidine treatment cycles, median	6	4.5	4.0	NA
Min-max	1–28	1.0–26.0	1.0–75.0	
Median overall survival, months	10.4	9.6	7.3	NA
Treatment discount. due to AE: Yes, n (%)	89 (37.0)	5 (3.4)	40 (5.2)	<0.0001
Neutropenia: Grade1–4, n (%)	71 (30.1)	42 (29.2)	296 (38.5)	0.0140
Grade 3–4	62 (26.3)	41 (28.5)	253 (32.9)	0.1226
Anemia: Grade 1–4, n (%)	48 (20.3)	30 (20.8)	459 (59.7)	<0.0001
Grade 3–4	37 (15.7)	29 (20.1)	305 (39.7)	<0.0001
Thrombopenia: Grade 1–4, n (%)	64 (27.1)	58 (40.3)	360 (46.8)	<0.0001
Grade 3–4	56 (23.7)	55 (38.2)	276 (35.9)	0.0011
Pyrexia: Grade 1–4, n (%)	89 (37.7)	32 (22.2)	209 (27.2)	0.0010
Grade 3–4	18 (7.6)	2 (1.4)	27 (3.5)	0.0043
Febrile neutropenia: Grade 1–4, n (%)	76 (32.2)	27 (18.8)	250 (32.5)	0.0030
Grade 3–4	66 (28.0)	27 (18.8)	250 (32.5)	0.0032
Pneumonia Grade 1–4, n (%)	57 (24.2)	39 (27.1)	165 (21.5)	0.2800
Grade 3–4	45 (19.1)	36 (25.0)	52 (6.8)	<0.0001
Upper resp. infection: Grade 1–4, n (%)	NR	NR	121 (15.7)	NA
Grade 3–4	NR	NR	9 (1.2)	
Urinary tract infection: Grade 1–4, n (%)	NR	NR	61 (7.9)	NA
Grade 3–4	NR	NR	6 (0.8)	
Nausea: Grade 1–4, n (%)	94 (39.8)	50 (34.7)	70 (9.1)	<0.0001
Grade 3–4	0 (0.0)	1 (0.7)	2 (0.3)	0.7020
Diarrhea: Grade 1–4, n (%)	87 (36.9)	48 (33.3)	76 (9.9)	<0.0001
Grade 3–4	0 (0.0)	4 (2.8)	4 (0.5)	0.0145
Constipation: Grade 1–4, n (%)	99 (41.9)	46 (38.9)	65 (8.5)	<0.0001
Grade 3–4	0 (0.0)	2 (1.4)	1 (0.1)	0.0610
Injection site reaction: Grade 1–4, n (%)	NR	NR	161 (20.9)	NA
Grade 3–4	NR	NR	8 (1.0)	
Fatigue: Grade 1–4, n (%)	54 (22.9)	24 (16.7)	251 (32.6)	<0.0001
Grade 3–4	0 (0.0)	2 (1.4)	33 (4.3)	0.0050

NA indicates not applicable; RR, relapsed or refractory; AZA, azacitidine; CCR, conventional care regimen; NR, not reported; ECOG-PS, Eastern Cooperative Oncology Group Performance Score, MRC, Medical Research Council.

## Data Availability

Data sharing of the data collected for the study is not planned. However, we are open to research questions asked by other researchers, and we are also open to data contributions by others. Participation requests or potential joint research proposals can be made at any timepoint to the corresponding author via email (dr.lisa.pleyer@gmail.com) and are subject to approval by the AGMT and its collaborators. The Ethics Committee Approval Votum, the study protocol, the signed Sponsor Approval Page, and the Informed Consent form of the AGMT and the Austrian Registry of Hypomethylating Agents can be made available.

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
