# Peer review of "Adverse Events in 1406 Patients Receiving 13,780 Cycles of Azacitidine within the Austrian Registry of Hypomethylating Agents—A Prospective Cohort Study of the AGMT Study-Group"

_cancers, 2022, doi:10.3390/cancers14102459_

Round 1

Reviewer 1 Report

The authors submitted a manuscript entitled “Adverse Events in 1406 Patients Receiving 13780 Cycles of Azacitidine within the Austrian Registry of Hypomethylating Agents - A prospective Cohort Study of the AGMT Study-Group” The manuscript is intriguing for the readers of “Cancers” because Azacitidine, alone and in combination with other agents are among new standard therapy for several hematological malignancies as discussed by the authors. However, there are several issues to be clarified.

Majors)

As discussed by the authors, the profile of adverse events of Azacitidine in this large survey in “Real World” was almost same as those of the pivotal clinical trials of this agent. What are the new findings in this study? Those should be described clearly or just describe “non” if not exist.

Minors)

Page 2, Line 86) “(iii) chronic myelomonocytic leukemia with <13·0 G/L white blood cell count and with ≥10% bone marrow blasts”

The sentence is unclear.  

Page 2, Line 93) “In patients with chronic myelomonocytic leukemia, approval was based on less than 20 patients included in myelodysplastic syndromes trials.”

The reference is nbetter to be added.

Page 6, Line 200) “Azacitidine was used as 1st line treatment in 838 (59.6%), as 2 nd line treatment in 301 patients (21.4%), and as ≥3rd line treatment in 267 (19.0%) of 1406 patients, respectively.”

Were there any difference in the profile of toxicities in relation to the line of treatment?

Page 6, Line 203) “Of all 1406 treated patients, 639 (45.4%) had an objective response, with 154 (11.0%) patients achieving a complete remission (CR), 54 (3.8%) a complete remission with incomplete marrow recovery (CRi), 84 (6.0%) a morphologic leukemia free state (MLFS), 25 (1.8%) a partial remission (PR) and 322 (22.9%) a hematologic improvement (HI) as best 206 response, respectively. Median (IQR) overall survival was 9.7 (3.8-18.8) months. The one- and three-year survival rates after azacitidine start were 49.2% and 17.9% for the total cohort, respectively. Further treatment characteristics and outcomes are listed in Supplemental Table S3.”

Were there any difference in prognostic factors of this study and those of the pivotal clinical trials?

Page 7, Line 236) “The cumulative effects of 122 adverse events (0.7%) occurring in 16 (0.09%) patients 236 resulted in ICU admission or were classified as life threatening. The cumulative effect of 237 256 adverse events (1.6%) led to a fatal outcome in 33 (2.3%) of 1406 patients.”

The frequency of grade 5 toxicity was apparently low. It is better to describe in detail and discuss the reason

Page 7, Line 239) “The adverse event duration was <3 days in 2677 (16.7%), 3-6 days in 4770 (29.7%), 1- 239 2 weeks in 3840 (23.9%), 2-3 weeks in 1593 (9.9%), 3-4 weeks in 914 (5.7%), and >4 weeks in 2210 (13.8%) adverse events of all 16023 documented adverse events, respectively. Grade 3-4 adverse events resolved within <3 days in 718 (11.4%), 3-6 days in 1606 (25.5%), 242 1-2 weeks in 1540 (24.5%), 2-3 weeks in 757 (12.0%), 3-4 weeks in 441 (7.0%), and >4 weeks in 1203 (19.1%) of all documented grade 3-4 adverse events, respectively.”

It is better to describe the characteristics of the adverse events, especially those of non-resolved ones.

Page 8, Figure 2) It is better to add the numbers of patients at risk under the Azacitidine cycle numbers.

Author Response

Comments and Suggestions for Authors:

The authors submitted a manuscript entitled “Adverse Events in 1406 Patients Receiving 13780 Cycles of Azacitidine within the Austrian Registry of Hypomethylating Agents - A prospective Cohort Study of the AGMT Study-Group” The manuscript is intriguing for the readers of “Cancers” because Azacitidine, alone and in combination with other agents are among new standard therapy for several hematological malignancies as discussed by the authors. However, there are several issues to be clarified.

Majors)

As discussed by the authors, the profile of adverse events of Azacitidine in this large survey in “Real World” was almost same as those of the pivotal clinical trials of this agent. What are the new findings in this study? Those should be described clearly or just describe “non” if not exist.

Author response: We thank the reviewer for this valuable point. As stated by the reviewer, we have discussed our results in the context of clinical trial data. On the one hand we were able to validate clinical trial data in the real world setting, on the other hand we complemented these data with additional findings. We have made these findings more clearly in the discussion and the conclusion:

494-497: “In addition, we report rates of skin- and mucosal infections, pain, bilirubin increase, GOT increase, GPT increase, and creatinine increase. These adverse events were not reported in any of the pivotal trials and therefore this registry analysis adds additional information to the safety profile of azacitidine”.

557-561: “Overall, the adverse event reporting in the Austrian Registry of Hypomethylating Agents was high and mostly comparable to that of published randomized clinical trials with some differences in frequency as outlined above. Furthermore, we have found adverse events not documented in published trials that complement published clinical trial data”.

Minors)

Page 2, Line 86) “(iii) chronic myelomonocytic leukemia with <13·0 G/L white blood cell count and with ≥10% bone marrow blasts”

The sentence is unclear.  

Author response: we have clarified this sentence

87-88: “myelodysplastic chronic myelomonocytic leukemia with a white blood cell count <13.0 G/L and with ≥10% bone marrow blasts”.

Page 2, Line 93) “In patients with chronic myelomonocytic leukemia, approval was based on less than 20 patients included in myelodysplastic syndromes trials.”

The reference is n better to be added.

Author response: we have added the exact patient numbers and the corresponding references.

93-94: “In patients with chronic myelomonocytic leukemia, approval was based on 6-14 patients included in myelodysplastic syndromes trials [3,20].”

Page 6, Line 200) “Azacitidine was used as 1st line treatment in 838 (59.6%), as 2 nd line treatment in 301 patients (21.4%), and as ≥3rd line treatment in 267 (19.0%) of 1406 patients, respectively.”

Were there any difference in the profile of toxicities in relation to the line of treatment?

Author response: we thank the reviewer for this valuable input. We found higher rates of anemia, thrombopenia and neutropenia in patients receiving 1st line treatment compared to 2nd and later lines of treatment. GI toxicity and infections were equally common in all treatment lines. We have added this information to the main text:

274-282: “Hematologic adverse events were more common in patients receiving azacitidine as 1st line treatment than in patients receiving azacitidine in later treatment lines. As such, grade 3-4 anemia was seen in 401 (47.8%) of 838 1st line patients and 209 (36.7%) of 568 ≥ 2nd line treated patients, respectively (p=0.0004). Grade 3-4 neutropenia was seen in 365 (43.5%) 1st line patients as compared to 143 (25.1%) ≥ 2nd line patients (p=<0.00001) and grade 3-4 thrombopenia was observed in 340 (40.5%) 1st line patients as compared to 177 (31.1%) ≥ 2nd line patients, respectively (p=0.0003). There was no difference in non-hematologic and infectious adverse events between 1st and later line treated patients (data not shown)”.

Page 6, Line 203) “Of all 1406 treated patients, 639 (45.4%) had an objective response, with 154 (11.0%) patients achieving a complete remission (CR), 54 (3.8%) a complete remission with incomplete marrow recovery (CRi), 84 (6.0%) a morphologic leukemia free state (MLFS), 25 (1.8%) a partial remission (PR) and 322 (22.9%) a hematologic improvement (HI) as best 206 response, respectively. Median (IQR) overall survival was 9.7 (3.8-18.8) months. The one- and three-year survival rates after azacitidine start were 49.2% and 17.9% for the total cohort, respectively. Further treatment characteristics and outcomes are listed in Supplemental Table S3.”

Were there any difference in prognostic factors of this study and those of the pivotal clinical trials?

Author response: We have added the following text in order to clarify this issue (please also see Tables 3 and 4):

386-389: “There were relevant differences regarding baseline and prognostic factors between the Austrian registry and the clinical trials. Gender distribution was significantly different across all trials. Patients in the Austrian registry had a significantly higher ECOG performance score, whereas IPSS prognostic score was higher in the clinical trials (Table 3).”

434-438: “There were relevant differences regarding baseline and prognostic factors between the Austrian registry and the clinical trials. Patients in the Austrian registry and the VIALE-A trial had a significantly higher ECOG performance score than patients in the AZA-AML-001 trial. MRC cytogenetic risk was higher in the clinical trials than in the Austrian registry (Table 4).”

Page 7, Line 236) “The cumulative effects of 122 adverse events (0.7%) occurring in 16 (0.09%) patients 236 resulted in ICU admission or were classified as life threatening. The cumulative effect of 237 256 adverse events (1.6%) led to a fatal outcome in 33 (2.3%) of 1406 patients.”

The frequency of grade 5 toxicity was apparently low. It is better to describe in detail and discuss the reason

We thank the reviewer for this input. We have added this information to the text and to the discussion as follows:

252-259: “The cumulative effects of 122 adverse events (0.7%) occurring in 16 (0.09%) patients resulted in ICU admission or were classified as life threatening. The cumulative effect of 256 adverse events (1.6%) led to a fatal outcome in 33 (2.3%) of 1406 patients. The fatal adverse events occurred after a median of 3 (IQR 2-8) azacitidine treatment cycles. Ten of 33 patients (30.3%) with a fatal adverse event experienced the adverse event in cycle 1-2 or > 7. At the timepoint of the fatal adverse event, patients had a median of 6 (IQR 4-10) adverse events. The most common were febrile neutropenia (32 adverse events total), pneumonia (24 adverse events) and sepsis (18 adverse events).”

539-546: “Grade 5 (fatal) adverse events were reported in 2.3% of all patients. In these patients, febrile neutropenia, pneumonia and sepsis were most commonly associated with a fatal outcome. We found a median of six adverse events at the time of death, indicating a multifactorial cause of death. The majority of fatal adverse events occurred either early (in the first two cycles) or later (cycle 7 and beyond). This is in line with clinical experience that patients are at highest risk for a fatal outcome when the underlying disease is either not yet or no longer controlled. Therefore, it is often hard to discriminate between drug side effects and the underlying disease as the cause of death.”

Page 7, Line 239) “The adverse event duration was <3 days in 2677 (16.7%), 3-6 days in 4770 (29.7%), 1- 239 2 weeks in 3840 (23.9%), 2-3 weeks in 1593 (9.9%), 3-4 weeks in 914 (5.7%), and >4 weeks in 2210 (13.8%) adverse events of all 16023 documented adverse events, respectively. Grade 3-4 adverse events resolved within <3 days in 718 (11.4%), 3-6 days in 1606 (25.5%), 242 1-2 weeks in 1540 (24.5%), 2-3 weeks in 757 (12.0%), 3-4 weeks in 441 (7.0%), and >4 weeks in 1203 (19.1%) of all documented grade 3-4 adverse events, respectively.”

It is better to describe the characteristics of the adverse events, especially those of non-resolved ones.

We thank the reviewer for this input. We have added more information about non-resolved/long lasting adverse events as follows:

265-267: “The most common adverse events lasting longer than 4 weeks were thrombopenia (391 events), febrile neutropenia (343 events), anemia (286 events) and fatigue (238 events), respectively.”

Page 8, Figure 2) It is better to add the numbers of patients at risk under the Azacitidine cycle numbers.

We have added the numbers of patients at risk to Figure 2 and 3.

Submission Date

02 April 2022

Date of this review

13 Apr 2022 09:58:56

Reviewer 2 Report

The current manuscript reports the adverse events in large cohort of Austrian patients associated with the treatment of myeloid neoplasia by hypomethylating agents particularly Azacytidine. This manuscript a well written and deserve to be published. Here are some of my suggestions, if included will certainly improve the quality of this manuscript.

  1. Majority of myeloid neoplasia is driven by some kind of somatic mutations, however this manuscript lacks any mention of the association of different kind of somatic mutations to the outcome of azacytidine treatment and associated adverse outcome. It will be very useful clinical information to analyze such association if there is any. Even absence of the correlation will be a useful information. There are several recent reports that suggest a certain molecular marker can be the predictive of hypomethylating agent treatment outcome. The mutational configuration of such a large Austrian cohort in itself will be a very good information and can be an independent manuscript, however a cursory closer look into the azacytidine associated adverse outcome with different mutations will be very important and clinically relevant.  
  2. A comparison of adverse events among decitabine treated cohort will be a good addition. I understand the fact that decitabine cohort is relatively smaller, still a comparison will be good, particularly with respect to hematologic adverse events.

Minor:

  1. The fonts and the sizes should consistent and similar throughout the manuscript.
  2. Figures are of very poor quality, they can be aesthetically improved to make better impression on readers.
  3. There are few punctuation error and therefore carefully proof read.

Author Response

Comments and Suggestions for Authors

The current manuscript reports the adverse events in large cohort of Austrian patients associated with the treatment of myeloid neoplasia by hypomethylating agents particularly Azacytidine. This manuscript a well written and deserve to be published. Here are some of my suggestions, if included will certainly improve the quality of this manuscript.

  1. Majority of myeloid neoplasia is driven by some kind of somatic mutations, however this manuscript lacks any mention of the association of different kind of somatic mutations to the outcome of azacytidine treatment and associated adverse outcome. It will be very useful clinical information to analyze such association if there is any. Even absence of the correlation will be a useful information. There are several recent reports that suggest a certain molecular marker can be the predictive of hypomethylating agent treatment outcome. The mutational configuration of such a large Austrian cohort in itself will be a very good information and can be an independent manuscript, however a cursory closer look into the azacytidine associated adverse outcome with different mutations will be very important and clinically relevant.  

We agree with the reviewer that associations with certain mutations and adverse events would be of high interest. Routine use of NGS analysis was implemented at different time points at the contributing centers. As such we have mutational data from roughly 2019 onward (in 173 patients, 12.3% of the total cohort). Therefore, the data set is not mature enough at this point in time to perform this analysis. We have added information regarding the mutational landscape of our cohort to the baseline characteristics:

183-186: “Data regarding the mutational landscape of the study cohort were available in 173 (12.3%) patients. Patients had a median of one mutation (IQR 1-2). The most common mutations included mutations in NPM1 (n=66), FLT3 (n=47), NF1 (n=27) and TET2 (n=22), respectively.”

  1. A comparison of adverse events among decitabine treated cohort will be a good addition. I understand the fact that decitabine cohort is relatively smaller, still a comparison will be good, particularly with respect to hematologic adverse events.

We thank the reviewer for this great suggestion. We have created a supplemental table 2 with data from the decitabine registry including base line characteristics and adverse events. Currently, patient numbers and data maturity do not allow a reliable comparison between decitabine and azacitidine treated patients. We are hoping to be able to provide these data in the future.

Minor:

  1. The fonts and the sizes should consistent and similar throughout the manuscript.

We have formatted the manuscript accordingly.

  1. Figures are of very poor quality, they can be aesthetically improved to make better impression on readers.

We agree with the reviewer and have corrected the figures according to the suggestions

  1. There are few punctuation error and therefore carefully proof read.

We have done our best to carefully proof read the manuscript and have corrected the mentioned errors.

Submission Date

02 April 2022

Date of this review

18 Apr 2022 16:58:3

Round 2

Reviewer 1 Report

I think the manuscript has been revised well and can be accepted.